# Comparison of Long-Term Pneumonia Risk between Spleen Injury and Non-Spleen Injury after Total Splenectomy—A Population-Based Study

**DOI:** 10.3390/jpm12020308

**Published:** 2022-02-18

**Authors:** Chun-Cheng Lin, Sheng-Der Hsu, Wu-Chien Chien, Chi-Hsiang Chung, Cheng-Jueng Chen, Chia-Ming Liang, Zhi-Jie Hong

**Affiliations:** 1Department of Surgery, Tri-Service General Hospital, National Defense Medical Center, Taipei 11490, Taiwan; ginnib00942@gmail.com; 2Division of Traumatology, Department of Surgery, Tri-Service General Hospital, National Defense Medical Center, Taipei 11490, Taiwan; doc20227@yahoo.com.tw (C.-J.C.); kevin.magic77@gmail.com (C.-M.L.); lgf670822@mail.ndmctsgh.edu.tw (Z.-J.H.); 3School of Public Health, National Defense Medical Center, Taipei 11490, Taiwan; chienwu@ndmctsgh.edu.tw; 4Department of Medical Research, Tri-Service General Hospital, National Defense Medical Center, Taipei 11490, Taiwan; g694810042@gmail.com

**Keywords:** splenectomy, pneumonia, spleen injury, splenectomized patients

## Abstract

Patients who undergo splenectomy are at a high risk of infection. We aimed to investigate the rate of pneumonia in patients who underwent splenectomy, specifically comparing those who had splenectomy due to spleen injury and those who had it for other reasons. A population-based cohort study was conducted. Overall, 17,498 patients who underwent splenectomy between 2000 and 2015 were enrolled, including 11,817 patients with a history of spleen injury and 5681 controls without spleen injury. The incidence of pneumonia was calculated at the end of 2016. A multivariable Cox proportional hazards regression model was used to compare the hazard ratio with 95% CI for pneumonia associated with the spleen injury-caused splenectomy and splenectomy due to other causes. The crude HR for patients with splenectomy due to spleen injury to develop pneumonia was 1.649. After adjusting for covariates, the adjusted hazard ratio was 1.567. There were statistically significant differences in all subgroups, except for the group with a tracking duration >10 years. We found an increase in pneumonia risk in the ‘spleen injury’ group when comparing it to that of the ‘other causes’ group, regardless of age, sex, and area of residence.

## 1. Introduction

The spleen is an organ associated with various immune functions. It is essential for the acute clearance of pathogens from the bloodstream. Patients who undergo splenectomy are at a significant risk of infection due to weakened phagocytosis [1], suppression of serum levels of immunoglobulin M [2], and changes in the environment where red blood cells clear solid waste. The risk of overwhelming infection is more than 50 times higher in post-splenectomy than in the general population [3]. This makes patients susceptible to pneumonia, which has a high mortality rate [4]. The spleen is commonly injured in blunt abdominal trauma; since removing the spleen in case of extensive injury is accepted in clinical practice, splenectomies are not uncommon surgeries. Other clinical scenarios may also require splenectomy [5].

Although the association between splenectomy and pneumonia has been previously studied [6,7,8], there has been no formal study examining the vulnerability to pneumonia among patients who underwent splenectomy due to trauma as compared to those who underwent splenectomy due to other indications. We conducted a nationwide retrospective population-based cohort study using claims data from the Taiwan National Health Insurance Program. We aimed to investigate the rate of pneumonia in patients with splenectomy for injury and that in patients with splenectomy due to other causes.

## 2. Materials and Methods

### 2.1. Data Sources

Data for this study were obtained from the National Health Insurance Research Database (NHIRD) of Taiwan. Taiwan’s National Health Insurance (NHI) program was implemented in 1995 and covers more than 99% of Taiwanese residents. After de-identification and anonymization, all records and original claims data used by the hospital for payment purposes are registered in the NHIRD. The database provides researchers with anonymous identification numbers associated with relevant claims information, including gender, date of birth, use of medical services, and prescriptions.

All hospitals and clinics in Taiwan claim that government payments depend on the NHI procedure code. To prevent improper use of medical resources, there is an independent peer review process to verify that the requirements of the procedure are reasonable based on medical records. This study was not subject to a comprehensive review by the Ethical Institution Review Committee of Tri-Service General Hospital.

### 2.2. Study Design

This was a retrospective cohort study. To create the case cohort, patients who underwent total splenectomy (International Classification of Diseases, Ninth Revision, Clinical Modification, ICD-9 procedure code 41.5) between 2000 and 2015 were identified and extracted from the health insurance database. We included patients who underwent total splenectomy between 2000 and 2015. We excluded those with total splenectomy before the index date, pneumonia before tracking, malignant neoplasm of the spleen, tuberculosis of the spleen, those who underwent radiography or chemotherapy, those aged less than 20 years, and those with unknown sex. Finally, we identified 17,498 patients who met the inclusion criteria. Patient information was reviewed chronologically until a diagnosis of pneumonia was made, or until 31 December 2015. The flowchart of study sample selection and design is detailed in Figure 1.

### 2.3. Outcome Measure

Study participant information was reviewed up from the index date until pneumonia (ICD-9-CM 480–486) was diagnosed according to radiological signs, clinical culture lab data, or identified by specialist in thoracic, infectious, or radiological medicine; until end of life; or until the tracking endpoint. We carried out an investigation to determine which group of patients with splenectomy, those whose indication was spleen injury or those whose indication was another condition, developed pneumonia.

### 2.4. Covariates

Sex, age groups (0–11, 12–19, 20–49, >50 y), education level (<12 y, >12 y), urbanization level of residence, and monthly income (in New Taiwan dollars; <18,000, 18,000–34,999, >35,000) were listed. The urbanization level of residence was divided into levels 1–4 based on the population size. Level 1 was defined as a population >1,250,000 individuals. Level 2 was defined as a population between 500,000 and 1,249,999. PerUrbanization levels 3 and 4 were defined as population between 149,999 and 499,999, and <149,999, respectively. Level of care was divided into hospital center, regional hospital, and local hospital. In Taiwan, a hospital center is defined as a hospital with more than 500 beds that has government approval after medical center evaluation and it can handle all emergent trauma at any time. Institutions on this level of care usually have the largest number of medical field specialists. Regional hospitals are defined as hospitals with more than 300 beds and government approval after regional hospital evaluation. It could manage trauma if a related specialist is on duty. Local hospitals are defined as hospitals with 20–99 beds. It could not handle most of the major trauma cases and needs to transferr them to another hospital.

### 2.5. Comorbidity

The Charlson Comorbidity Index (CCI) was used to assess comorbidities on a scale of 0–4. A score of zero indicates that no comorbidities were found, and a higher score indicated a higher burden of comorbidities. The index uses the ICD-9-CM code to classify comorbidities, score each comorbidity category, and combine all scores to calculate a single comorbidity score. The CCI was developed in 1987 as a prognostic classification method [9,10]. It was originally developed to explain the impact of adverse medical conditions on patients in longitudinal studies and has been validated in many clinical settings [11,12].

### 2.6. Statistical Analysis

All analyses were performed using SPSS software for Windows version 19.0.a c2 and *t*-tests were used to evaluate the differences in demographic data and comorbidities between the splenectomy due to spleen injury and due to other causes. Fisher’s exact test for categorical variables was used to statistically examine the differences between the two cohorts. Each baseline characteristic was viewed as a distinct dichotomous variable. Those with a *p* value <0.001 were determined to be potentially significant covariates in further analyses. For the analysis of competing risks, Fine and Gray’s survival analysis was used to calculate the competing risk hazard ratio (HR) (adjusted for the variables listed in Table 1), and the incidence of psychiatric disorders or death were treated as competing risks. To examine the effect size, adjusted HRs and 95% confidence intervals (95% CIs) were calculated. A two-tailed statistical test was performed, and a *p* value < 0.05 was considered significant. Fine and Gray’s survival analysis was conducted using R 2.15.1 and R integration package for IBM SPSS (STATS_COMPRISK).

## 3. Results

In this study, we initially considered 20,355 patients who underwent splenectomy. After exclusion, 17,498 patients remained and were included. The cohort comprised of 11,817 people who have had splenectomy due to spleen injury and 5681 who have had it for other causes, as shown in Figure 1. The minimum follow-up duration was 0.01 years and the maximum follow-up period was 15.76 years. Of those who underwent splenectomy due to injury, 1917 developed pneumonia, as compared to 720 patients who developed pneumonia after splenectomy due to other causes.

Table 1 shows the sex, age, monthly insurance premiums, comorbidities, urbanization level, and level of care of patients with splenectomy at the beginning of enrollment. There were some statistical differences between the two groups, including sex, age, comorbidity, urbanization, and level of care. Patients with spleen injury were less likely to be males, more likely to be middle-aged (45–64 years), tended to have higher CCI scores, had a higher proportion of residents living in areas with urbanization levels 1 or 2, were hospitalized in medical centers, and were living in Northern Taiwan. Table 2 shows that patients with splenectomy due to injury and due to other causes are divided into those having developed pneumonia or not at the endpoint. The crude HR for patients with splenectomy due to spleen injury to develop pneumonia was 1.649.

There was also a significant difference in the risk of pneumonia between the splenectomy due to injury cohort and the splenectomy due to other causes cohort over the 15-year follow-up period (log-rank test: *p* < 0.001) (Figure 2).

The analysis of factors associated with pneumonia stratified by variables including sex, age, insurance premium, season of developing pneumonia, urbanization residence and level of care after pneumonia using Cox regression (Appendix A). The adjusted hazard ratio of pneumonia for patients with splenectomy due to spleen injury and elective splenectomy is 1.567. (1.567 CI 1.427–1.721, *p* < 0.001). The hazard ratio of pneumonia for patients with splenectomy due to spleen trauma and elective splenectomy in other subgroups like sex, age, season of developing pneumonia, urbanization residence and level of care are more than 1 and there are significant differences between the 2 groups of patients with splenectomy, due to injury or due to other causes. Except for a subgroup of insured premium patients, there was no statistical difference between patients with income less than 18,000 and those with an income higher than 35,000.

Table 3 shows the stratified analysis of different tracking periods using Cox regression. There were statistically significant differences in all subgroups, except in the group with a tracking period ≥10 years.

## 4. Discussion

This is the first population-based study to compare the risk of pneumonia among patients who underwent total splenectomy due to different indications. There is an increased risk of pneumonia after splenectomy, regardless of the medical indication for splenectomy [13]; however, no study has assessed whether different indications of splenectomy would lead to different infection risks.

In our study, the overall incidence of pneumonia was 1.297-fold higher in the spleen injury group than in the other causes group. We also found that the incidence of pneumonia was higher in patients who underwent splenectomy due to spleen injury regardless of sex, age and level of care. Patients in the spleen injury group had a higher CCI and tended to be older than those in the other causes group. This suggests that comorbidity and age are related to the incidence of spleen injury. The spleen injury group had a higher CCI because most patients in the group were older people, and they had poor physical performance and were unable to avoid trauma incidents; however, the disease that required total splenectomy in the other causes group usually develops at a young age.

As in previous studies [5,14,15,16], the incidence of pneumonia in the first year after total splenectomy was the highest regardless of the presence/absence of spleen injury, and the risk declined over time. Both the crude and adjusted hazard ratios gradually decreased with increasing tracking time in our study. We noticed that both the groups with splenectomy due to injury or due to other causes had high incidence of pneumonia (103.39 versus 67.8 per 1000 person-years). However, when the tracking time was more than 10 years, the adjusted hazard ratio for pneumonia was not significantly different (*p* = 0.072). This suggested that the effects of spleen injury reduced over time.

Multiorgan injury after a severe trauma incident can induce systemic inflammatory responses, and lung contusion is a risk factor for further infection [17]. In a retrospective study enrolling 1044 ventilated trauma patients in United State [18], rib fractures, pulmonary contusion, spinal cord injury, aspiration, traumatic brain injury and failed prehospital intubation were found to be significant predictors of pneumonia. Brain injury might induce several complications, including altered mental status, dysphagia, inability to clear secretion, and impaired cough function. These are risk factors for pneumonia after trauma incidents [19]. Another population-based study in Taiwan [20] found that patients after spleen injury have significantly higher long-term pneumonia occurrences among post splenectomy groups. In that study, outcomes of patients with splenectomy were compared with those of patients whose spleen was preserved. The incidence of long-term pneumonia was also significantly higher in the patients who underwent splenectomy. We investigated another subgroup to compare patients after splenectomy due to spleen trauma with and without chest injury. (Appendix A) We found that there is no significant different between pneumonia occurrence in the groups whose injury was to the chest or otherwise (Appendix A).

Previous studies have revealed that splenectomy performed for a hematological disease, with or without malignancy, has a higher risk for pneumonia than splenectomy performed due to spleen injury [21,22]. However, our data revealed that those in the other causes group had lower risks than those in the other group. Our findings show that splenectomy for hematological diseases does not have additional risk of pneumonia when compared to splenectomy for spleen injury.

A prospective cohort study in Cedars-Sinai Medical Center [23] showed no differences between traumatic and elective splenectomy regarding to overall infectious complications. However, sample sizes in the study were small and only early postoperative data were assessed.

Vaccination, including influenza and streptococcus vaccine after total splenectomy, are suggested to decrease the risk of pneumonia [24]. Vaccination does not only reduce the risk of pneumonia, but also reduces medical cost [25]. In our study, we did not have access to data regarding the effect of vaccination.

This study had some limitations. First, in this type of claims database research, records may not include all the data. In addition, these records may not reflect the fact that many different healthcare professionals are involved in patient care, so the measurement of risk factors and outcomes in the entire database may not be as accurate and consistent as that achieved by a prospective cohort study design. Second, we could not know exactly which type of pneumonia, viral or bacterial, occurred. Third, we also were not able to know what kind of antibiotics were used after pneumonia, or the length of antibiotic administration in both inpatient or out-patient settings. Another limitation is the extremely chronological heterogeneity in our cohort regarding the period during which patients developed pneumonia. Another limitation is that we could not determine the patients’ immunization status from the NHIRD because Taiwan’s NHI vaccination with pneumococcal vaccine does not consider splenectomy status; patients must pay through commercial insurance or at their own expense.

## 5. Conclusions

In conclusion, we found an increase in the risk of pneumonia in the spleen injury group, compared with the other causes group, regardless of age, sex, urbanization of residence, level of care.

## Figures and Tables

**Figure 1 jpm-12-00308-f001:**
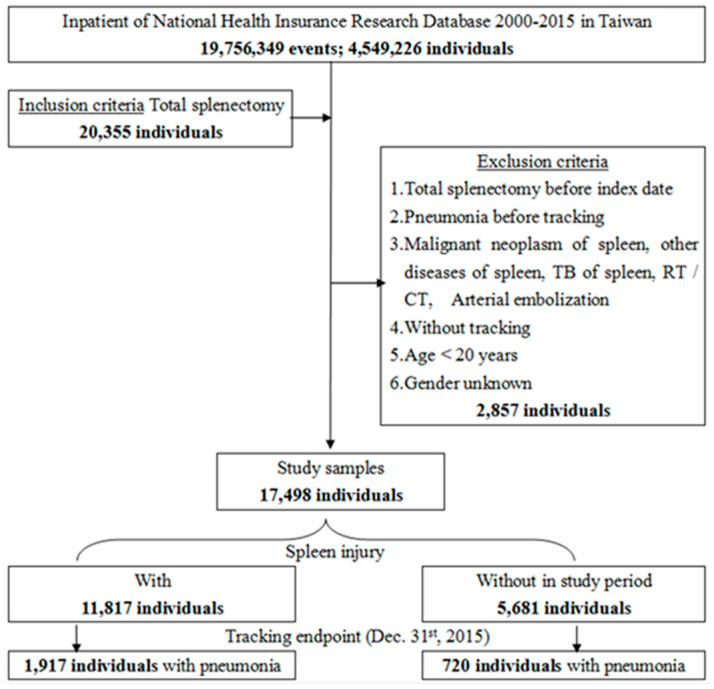
The flowchart of study sample selection and design.

**Figure 2 jpm-12-00308-f002:**
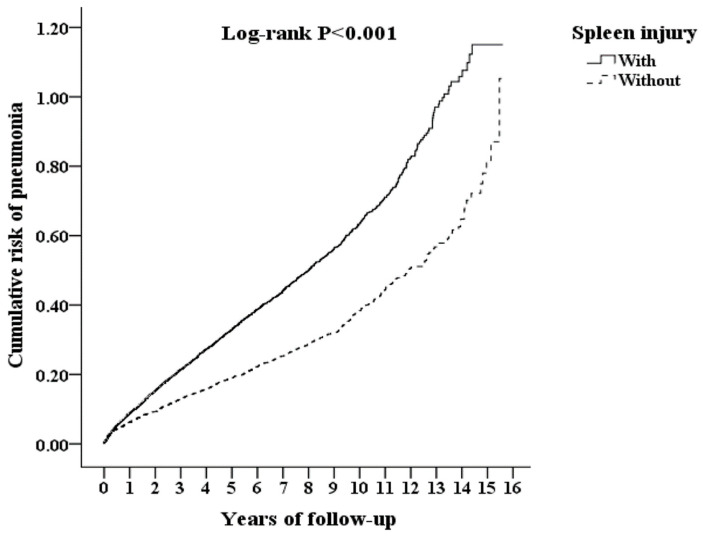
Kaplan–Meier estimate of the cumulative risk of pneumonia among study participants according to baseline spleen injury status.

**Table 1 jpm-12-00308-t001:** Baseline characteristics of study participants.

Variables	All Participants	Splenectomy Due to Injury	Splenectomy Due to Other Causes	*p*
Total, *n* (%)	17,498 (100.00)	11,817(67.53)	5681 (32.47)	
Sex, (*n* (%)				<0.001
Male	10,687 (61.08)	6589 (55.76)	4098 (72.14)	
Female	6811 (38.92)	5228 (44.24)	1583(27.86)	
Age (years), mean ± SD	52.55 ± 16.98	57.39 ± 15.01	42.48 ± 16.43	<0.001
Age group (years), *n* (%)				
20–44	5913(33.79)	2533 (21.44)	3380 (59.50)	
45–64	6786 (38.78)	5165 (43.71)	1621 (28.53)	
≥65	4799 (27.43)	4119 (34.86)	680 (11.97)	
Insurance premium (NT$), *n* (%)				0.279
<18,000	17,147 (97.99)	11,580 (97.99)	5567(97.99)	
18,000–34,999	268 (1.53)	175 (1.48)	93 (1.64)	
≥35,000	83 (0.47)	62 (0.52)	21 (0.37)	
CCI, mean ± SD	1.27 ± 2.57	1.78 ± 2.93	0.20 ± 0.89	<0.001
Urbanization level, *n* (%)				<0.001
1 (highest)	6884 (39.34)	5443 (46.06)	1441 (25.37)	
2	7787 (44.50)	5153 (43.61)	2634 (46.37)	
3	1119 (6.40)	496 (4.20)	623 (10.97)	
4 (lowest)	1708 (9.76)	725 (6.14)	983 (17.30)	
Level of care, *n* (%)				<0.001
Hospital center	9933 (56.77)	8055 (68.16)	1878 (33.06)	
Regional hospital	6397 (36.56)	3422 (28.96)	2975 (52.37)	
Local hospital	1168 (6.68)	340 (2.88)	828 (14.57)	

The *p*-values are based on chi-square tests or Fisher’s exact test for categorical variables and *t*-tests for continuous variables. CCI, Charlson’s Comorbidity Index.

**Table 2 jpm-12-00308-t002:** Patients with splenectomy due to injury and due to other causes divided into having developed pneumonia or not.

Variables	Total	Splenectomy Due to Injury	Splenectomy Not Due to an Injury	*p*
Total, *n* (%)	17,498 (100.00)	11,817 (67.53)	5681 (32.47)	
Pneumonia, *n* (%)				<0.001
Without	14,861 (84.93)	9900 (83.78)	4961 (87.33)	
With	2637 (15.07)	1917 (16.22)	720 (12.67)	

The *p*-values are based on chi-square tests or Fisher’s exact test for categorical variables and *t*-tests for continuous variables. CCI, Charlson’s Comorbidity Index.

**Table 3 jpm-12-00308-t003:** Risk factors for pneumonia according to spleen injury status and the duration of follow-up, and the effect of spleen injury on the risk of pneumonia.

	Splenectomy Due to Injury			Splenectomy Due to Other Causes				Due to Spleen Injury vs. Due to Other Causes (Reference)	
	Events	PYs	Rate (per 1000 PYs)	Events	PYs	Rate (per 1000 PYs)	Ratio	Adjusted HR (95% CI)	*p*
Overall	1917	123,167	15.56	720	60,004	12.00	1.297	1.567 (1.427–1.721)	<0.001
Follow-up period (years)									
<1	673	6509	103.39	204	3009	67.80	1.525	1.842 (1.670–2.035)	<0.001
1–2.9	540	22,182	24.34	157	7305	21.49	1.133	1.368 (1.247–1.514)	<0.001
3–4.9	283	22,333	12.67	101	8001	12.62	1.004	1.214 (1.105–1.338)	<0.001
5–9.9	323	48,632	6.64	176	24,120	7.30	0.910	1.101 (1.003–1.209)	0.045
≥10	98	23,512	4.17	82	17,570	4.67	0.893	1.078 (0.982–1.125)	0.072

The adjusted HRs were calculated using Cox proportional hazards regression with all the variables listed in the Table 3 included in the model. CI, confidence interval; HR, hazard ratio; PYs, person-years.

## Data Availability

All of our manuscript are available.

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
