# Peer review of "Comparison of Long-Term Pneumonia Risk between Spleen Injury and Non-Spleen Injury after Total Splenectomy—A Population-Based Study"

_jpm, 2022, doi:10.3390/jpm12020308_

Round 1

Reviewer 1 Report

I would like to congratulate the authors on such a large study with wonderful follow-up.  Quite an undertaking. 

My main concerns, which are also listed below, is what this paper adds to the scientific literature.  I don't know if this is at all practice changing, or adds relevant information in regards to this patient population.  

I have some suggestions to help strengthen the paper, which are primarily around word phrasing and data presentation. 

Abstract:

1) I am confused by the naming of the groups "spleen injury" and "non-spleen injury" groups.  The wording would be better if it was something perhaps : "We aimed to investigate the rate of pneumonia in splenectomy patients and specifically those who had splenectomy due to injury and those with a splenectomy not due to an injury"  Just try and make the group distinction clear.

2) "The adjusted hazard ratio for the spleen injury and non-spleen injury groups revealed that a higher hazard ratio indicated a shorter tracking period"  Confusing sentence.  Perhaps delete?  You comment on the meaningful differences in the next sentence and comment upon your main outcome after that. 

INTRODUCTION

1) I feel some references are missing.  "...including pneumonia, which has a high mortality rate (reference?)"

2) The last sentence of the first paragraph doesn't seem to fit.  Consider re-wording: example "While an important immunological organ, the spleen is commonly injured in blunt abdominal trauma, and splenectomies are not uncommon surgeries, for numerous reasons (reference)".  Or consider moving the last sentence of the first paragraph to the second paragraph, where the authors discuss splenectomy.

2) last sentence - again, consider re-wording for clarification of the two groups, as mentioned in the abstract

MATERIALS & METHODS

general - how was the information from the database extracted?  Is this a searchable database that exported the information that was searched for or did some people have to extract this data?  Was the data extracted by a single person? Verified by a second author?

1) consider changing the wording in the section 2.2 Study Design.  line 65 "In total, 17,498 enrolled patients...."  Perhaps better to say what the inclusion criteria are, and what the exclusion criteria are, and then say "we identified 17498 patients who met inclusion criteria"

2) Outcome measure - please better define your primary outcome, and comment that it was whether the patient developed pneumonia, not until "occurance of pneumonia"  because not everyone got a pneumonia.

3) line 75 - please clarify.  your investigated whether the patients had pneumonia earlier - earlier to what?  before splenectomy? 

RESULTS

line 108 - you assessed 20355 patients that met inclusion criteria, but after exclusion you ended up with 17498 patients.   Please re-word to clarify your included patients. 

table 1 - level of care: is this where the splenectomy was done?  Is this their first point of contact with health care?  please explain what the level of care represents.

table 1 - season.  Does this need to be in the table?  Could just report the values in the text?  Does the season of operation matter?  I could see perhaps being more interested in the season of when they get a pneumonia, but I don't feel this adds anything. 

table 2 title - confusing.  reword please.  

table 2 - sex, age, insurance premium, CCI, season, and urbanization are the same from Table 1.  why are they repeated?  Unnecessary.  I would suggest to remove all repeated values.  Perhaps make table 2 just the pneumonia data or add it into table 1

table 2 - why is the season numbers different between table 1 and 2? If this is the season of the pneumonia, I don't see that written anywhere, and needs to be clear.

DISCUSSION

line 168-169: I don't think you show that the incidence of pneumonia was higher in patients who underwent splenectomy.  Your population was all patients with splenectomy, you didn't have a non-splenectomy comparison group. 

Some more discussion in regards to the authors thoughts as why the group that had a splenectomy after a spleen injury would be more prone to pneumonia would is needed.  There are comments about the acute phase after the initial injury/trauma and splenectomy, but from Figure 2, the acute phase (say < 1 year) doesn't show a large change. 

GENERAL

I find tables 3 and 4 confusing and difficult to interpret.  Perhaps include as supplemental information?  

If available, ensure all grammar has been checked by English speaking proof reader.  I wonder if some of my misunderstanding is based on small grammar errors. 

I also struggle somewhat to see how this adds a lot to the literature.  I agree this large population study was a wonderful undertaking, but the vaccination is important and suggested to all splenectomy patients.  If vaccination status was known, that would be useful.  If there was a comparison group that didn't have splenectomy performed, that would also be very interesting (perhaps trauma patients who didn't have their spleen removed?

These next three comments might just be outside the scope of the database that was used in this manuscript.
How was the pneumonia diagnosed?  Did everyone have chest xrays, or in some cases would a physician have started treatment only based on symptoms?

Was there any way to distinguish viral vs bacterial pneumonia based on the database that was searched?

Was the pneumonia more severe in one population ?  For example, did one group require more IV antibiotics and admission vs out patient treatment?  

Author Response

Q1. Abstract:

  1. I am confused by the naming of the groups "spleen injury" and "non-spleen injury" groups.  The wording would be better if it was something perhaps: "We aimed to investigate the rate of pneumonia in splenectomy patients and specifically those who had splenectomy due to injury and those with a splenectomy not due to an injury" Just try and make the group distinction clear.

Authors

Thank for your suggestion. We have corrected the sentence on page 2 line 45-47, as well as the wording in the remainder of the manuscript.

  1. "The adjusted hazard ratio for the spleen injury and non-spleen injury groups revealed that a higher hazard ratio indicated a shorter tracking period" Confusing sentence.  Perhaps delete?  You comment on the meaningful differences in the next sentence and comment upon your main outcome after that.

Authors

   We appreciate your suggestions on improving our manuscript. After discussion, we decided to delete the sentence. Thank you.

Q2. INTRODUCTION

  1. I feel some references are missing.  "...including pneumonia, which has a high mortality rate (reference?)"

Authors

Thank for your suggestion. We have added a reference on page 1, line 36. Thank you.

  1. The last sentence of the first paragraph doesn't seem to fit.  Consider re-wording: example "While an important immunological organ, the spleen is commonly injured in blunt abdominal trauma, and splenectomies are not uncommon surgeries, for numerous reasons (reference)".  Or consider moving the last sentence of the first paragraph to the second paragraph, where the authors discuss splenectomy.

Authors

We have reworded the sentence based on your suggestion. Please see page 1 line 36-39. Thank you.

  1. last sentence - again, consider re-wording for clarification of the two groups, as mentioned in the abstract

Authors

We have reworked the sentence for clarity. page 1 line 14-16. Thank you.

Q3. MATERIALS & METHODS

  1. general - how was the information from the database extracted? 

Authors

Dr. Chi-Hsiang Chung, who specializes in big data analysis and is especially experienced in National Health Insurance Research Database (NHIRD) of Taiwan, extracted the information.

  1. Is this a searchable database that exported the information that was searched for or did some people have to extract this data?  Was the data extracted by a single person? Verified by a second author?

Authors

Prof. Wu-Chien Chien, who is a specialist in big data analysis as well, verified the information. He is also experienced in the medium of the National Health Insurance Research Database (NHIRD) of Taiwan, and is the head of the department responsible for big data analysis in a tertiary medical center.

  1. consider changing the wording in the section 2.2 Study Design.  line 65 "In total, 17,498 enrolled patients...."  Perhaps better to say what the inclusion criteria are, and what the exclusion criteria are, and then say "we identified 17498 patients who met inclusion criteria"

Authors

We appreciate your suggestion, and have employed it. page 2 line 66-70. Thank you.

  1. Outcome measure - please better define your primary outcome, and comment that it was whether the patient developed pneumonia, not until "occurance of pneumonia" because not everyone got a pneumonia.

line 75 - please clarify.  your investigated whether the patients had pneumonia earlier - earlier to what?  before splenectomy? 

Authors

We appreciate your suggestions for improving our manuscript. We have reworded the sentence to enhance clarity. Please see page 3 line 77-79. Thank you.

Q4. RESULTS

  1. line 108 - you assessed 20355 patients that met inclusion criteria, but after exclusion you ended up with 17498 patients.   Please re-word to clarify your included patients.

 Authors

Thank you for your suggestions. We have re-worded the sentence to clarify the meaning. Please see page 4 line 122-124.

  1. table 1 - level of care: is this where the splenectomy was done?  Is this their first point of contact with health care?  please explain what the level of care represents.

Authors

Your suggestions for improving our manuscript are much appreciated. The variable of level of care in table 1 means where splenectomy was done and it also their first point of contact with health care. We have added definitions for level of care in Taiwan in the MATERIALS & METHODS section. Please see page 3 line 90-97. Thank you.

  1. table 1 - season.  Does this need to be in the table?  Could just report the values in the text?  Does the season of operation matter?  I could see perhaps being more interested in the season of when they get a pneumonia, but I don't feel this adds anything. 

Authors

Thank you for the valuable comment. We have deleted the value.

  1. table 2 title - confusing.  reword please.  

table 2 - sex, age, insurance premium, CCI, season, and urbanization are the same from Table 1.  why are they repeated?  Unnecessary.  I would suggest to remove all repeated values.  Perhaps make table 2 just the pneumonia data or add it into table 1

table 2 - why is the season numbers different between table 1 and 2? If this is the season of the pneumonia, I don't see that written anywhere, and needs to be clear.

Authors

We have revised our manuscript and deleted the non-essential information. We have re-worded the title of table 2 per your suggestion. Please check page 6 Table.2. Thank you.

Q5. DISCUSSION

  1. line 168-169: I don't think you show that the incidence of pneumonia was higher in patients who underwent splenectomy.  Your population was all patients with splenectomy, you didn't have a non-splenectomy comparison group. 

Authors

We really appreciate your comment.  We have revised our manuscript and correct it in the literature. Please see the page7 line181 and Table. 3 Thank you.

  1. Some more discussion in regards to the authors thoughts as why the group that had a splenectomy after a spleen injury would be more prone to pneumonia would is needed. 

Authors

We really appreciate your suggestions. We conducted a literature search for articles that may help us discuss this further. We have added these details to page7-8 line198-212. Thank you.

  1. There are comments about the acute phase after the initial injury/trauma and splenectomy, but from Figure 2, the acute phase (say < 1 year) doesn't show a large change.

Authors

Your suggestions for our manuscript are appreciated. A significant difference does exist between both groups, as shown in figure 2. The same result is displayed in table 3.

Q6.

  1. I find tables 3 and 4 confusing and difficult to interpret.  Perhaps include as supplemental information?  

Authors

We really appreciate your suggestions in reviewing our manuscript. We would remove table 3 and move table 4 as supplemental table S1, and we still mention the results in the literature.

  1. If available, ensure all grammar has been checked by English speaking proof reader.  I wonder if some of my misunderstanding is based on small grammar errors. 

Authors

We have had our manuscript checked for language and grammar by editage.

We have attached the English-language editing certificate along with our manuscript for your perusal. Thank you.

  1. I also struggle somewhat to see how this adds a lot to the literature.  I agree this large population study was a wonderful undertaking, but the vaccination is important and suggested to all splenectomy patients.  If vaccination status was known, that would be useful.  If there was a comparison group that didn't have splenectomy performed, that would also be very interesting (perhaps trauma patients who didn't have their spleen removed?

Authors

Data on vaccination status is not available in the National Health Insurance Research Database (NHIRD) of Taiwan, which is a limitation of the database. We have mentioned this as a limitation in our study in page 8 line 236-238.

  1. These next three comments might just be outside the scope of the database that was used in this manuscript.
    How was the pneumonia diagnosed?  Did everyone have chest x-rays, or in some cases would a physician have started treatment only based on symptoms?

Authors

The diagnosis of pneumonia (ICD-9-CM 480–486) was according to related images, clinic culture lab data or identified by specialist in thoracic, infection medicine or radiologist. Please see page 3 line 77-79. Thank you.

  1. Was there any way to distinguish viral vs bacterial pneumonia based on the database that was searched?

Authors

This is a limitation of the database. We could not distinguish viral and bacterial pneumonia as this was not mentioned in the NHIRD. We have mentioned this as a limitation in the discussion section of the manuscript. Please see page 9 line 231-232.

Was the pneumonia more severe in one population?  For example, did one group require more IV antibiotics and admission vs out-patient treatment?  

Authors

As specified previously, this is a limitation of the database. Treatment, clinical severity, as well as the etiology of the pneumonia could not be determined using the NHIRD. We have mentioned this in the discussion section of the manuscript. Please see page 9 line 232-234.

Reviewer 2 Report

Good written paper, I have some remarks: date of observation for pneumonia should be extended, because within the study there are some patients with follow up of 15 years and some with follow up of 0,01 year - extremely heterogenous group regarding this. Another big concern is, that you have no data about vaccination, as you already mentioned in the discussion. In the paper the reasons for having more pneumonia could have been more clearly pointed out. Because of politrauma etc. one would clearly expect, that in the early postoperative period of 3 months, there would be higher incidence of pneumonia in the injury group. To see the impact of splenectomy on pneumonia probably all these early events should not be included. Or at least you could compare the results including this group of patients with the results excluding them. 

Author Response

  1. because within the study there are some patients with follow up of 15 years and some with follow up of 0,01 year - extremely heterogenous group regarding this.

Authors

We agree with you comment. We have mentioned this as a limitation in the manuscript. See page 9 line 234-236.

  1. Another big concern is, that you have no data about vaccination, as you already mentioned in the discussion.

Authors

It is outside the scope of the National Health Insurance Research Database (NHIRD) of Taiwan. We could not determine the vaccination status of the patients. We have mentioned this as a limitation. Please see page 9 line 236-238.

  1. In the paper the reasons for having more pneumonia could have been more clearly pointed out. Because of politrauma etc.

Authors

We investigated another subgroup to compare splenectomy due to spleen injury with and without lung injury (ICD-9-CM 860, 862 and 875). The result of the investigation is following:

Table S2. Factors of pneumonia after splenectomy by using Cox regression

Groups

Population

Events

Events %

Rate (per 1,000PYs)

Adjusted HR

P value

Not due to spleen injury

5,681

720

12.67

12.00

Reference

Due to spleen injury

11,817

1,917

16.22

15.56

1.567

<0.001

   With spleen injury, without chest injury

11,665

1,887

16.18

15.50

1.448

<0.001

   With spleen injury, with chest injury

152

30

19.74

21.04

2.241

<0.001

PYs = Person-years; Adjusted HR = Adjusted Hazard ratio: Adjusted for the variables listed in Table 3.; CI = confidence interval                                                                  

Table S3. pneumonia with / without chest injury among spleen injury by using Cox regression

Groups

Adjusted HR

P value

   With spleen injury, without chest injury

Reference

   With spleen injury, with chest injury

1.351

0.603

According the results, we found that, within the traumatic spleen injury subgroup, there is a significantly increased risk of pneumonia in patients whose splenectomy was due to chest injury when compared to the ‘other causes’ group. However, there was no significant increase in risk when comparing the group with traumatic injury to the chest and the group with traumatic injury elsewhere. (P value= 0.603). I would add the result in page 8 line207-212.

  1. One would clearly expect, that in the early postoperative period of 3 months, there would be higher incidence of pneumonia in the injury group. To see the impact of splenectomy on pneumonia probably all these early events should not be included. Or at least you could compare the results including this group of patients with the results excluding them. 

Authors

We investigated another subgroup with pneumonia after splenectomy to compare the tracking period less than 3 months and more than 3 months. The result is as following table S4. There is higher incidence of pneumonia in the injury group in the early postoperative period of 3 months. We also found that the group of tracking period more than 3 months had a higher hazard ratio than group of tracking period less than 3 months.

Table S4. Factors of pneumonia among different tracking period by using Cox regression

Due to spleen injury vs. not due to spleen injury (reference)

Tracking period (months)

Adjusted HR1 (95% CI)

95% CI

95% CI

P

<3

1.523

1.406

1.708

<0.001

≧3

2.596

2.050

2.911

<0.001

PYs = Person-years; Adjusted HR = Adjusted Hazard ratio: Adjusted for the variables listed in Table S1.; CI = confidence interval

Round 2

Reviewer 1 Report

thank you for your quick efforts with the manuscript, and for addressing my main concerns.  well done.